# Integration of Radiometric Ground-Based Data and High-Resolution QuickBird Imagery with Multivariate Modeling to Estimate Maize Traits in the Nile Delta of Egypt

**DOI:** 10.3390/s21113915

**Published:** 2021-06-06

**Authors:** Adel H. Elmetwalli, Andrew N. Tyler, Farahat S. Moghanm, Saad A.M. Alamri, Ebrahem M. Eid, Salah Elsayed

**Affiliations:** 1Agricultural Engineering Department, Faculty of Agriculture, Tanta University, Tanta 31527, Egypt; adel.elmetwali@agr.tanta.edu.eg; 2School of Biological and Environmental Sciences, University of Stirling, Stirling, Scotland FK9 4LA, UK; a.n.tyler@stir.ac.uk; 3Soil and Water Department, Faculty of Agriculture, Kafrelsheikh University, Kafr El-Sheikh 33516, Egypt; fsaadr@yahoo.ca; 4Biology Department, College of Science, King Khalid University, Abha 61321, Saudi Arabia; saralomari@kku.edu.sa; 5Botany Department, Faculty of Science, Kafrelsheikh University, Kafr El-Sheikh 33516, Egypt; 6Agricultural Engineering, Evaluation of Natural Resources Department, Environmental Studies and Research Institute, University of Sadat City, Minufiya 32897, Egypt; salah.emam@esri.usc.edu.eg

**Keywords:** maize, QuickBird imagery, spectral indices, water stress, salinity stress, PLSR and MLR

## Abstract

In site-specific management, rapid and accurate identification of crop stress at a large scale is critical. Radiometric ground-based data and satellite imaging with advanced spatial and spectral resolution allow for a deeper understanding of crop stress and the level of stress in a given area. This research aimed to assess the potential of radiometric ground-based data and high-resolution QuickBird satellite imagery to determine the leaf area index (LAI), biomass fresh weight (BFW) and chlorophyll meter (Chlm) of maize across well-irrigated, water stress and salinity stress areas in the Nile Delta of Egypt. Partial least squares regression (PLSR) and multiple linear regression (MLR) were evaluated to estimate the three measured traits based on vegetation spectral indices (vegetation-SRIs) derived from these methods and their combination. Maize field visits were conducted during the summer seasons from 28 to 30 July 2007 to collect ground reference data concurrent with the acquisition of radiometric ground-based measurements and QuickBird satellite imagery. The results showed that the majority of vegetation-SRIs extracted from radiometric ground-based data and high-resolution satellite images were more effective in estimating LAI, BFW, and Chlm. In general, the vegetation-SRIs of radiometric ground-based data showed higher R^2^ with measured traits compared to the vegetation-SRIs extracted from high-resolution satellite imagery. The coefficient of determination (R^2^) of the significant relationships between vegetation-SRIs of both methods and three measured traits varied from 0.64 to 0.89. For example, with QuickBird high-resolution satellite images, the relationships of the green normalized difference vegetation index (GNDVI) with LAI and BFW showed the highest R^2^ of 0.80 and 0.84, respectively. Overall, the ground-based vegetation-SRIs and the satellite-based indices were found to be in good agreement to assess the measured traits of maize. Both the calibration (Cal.) and validation (Val.) models of PLSR and MLR showed the highest performance in predicting the three measured traits based on the combination of vegetation-SRIs from radiometric ground-based data and high-resolution QuickBird satellite imagery. For example, validation (Val.) models of PLSR and MLR showed the highest performance in predicting the measured traits based on the combination of vegetation-SRIs from radiometric ground-based data and high-resolution QuickBird satellite imagery with R^2^ (0.91) of both methods for LAI, R^2^ (0.91–0.93) for BFW respectively, and R^2^ (0.82) of both methods for Chlm. The models of PLSR and MLR showed approximately the same performance in predicting the three measured traits and no clear difference was found between them and their combinations. In conclusion, the results obtained from this study showed that radiometric ground-based measurements and high spectral resolution remote-sensing imagery have the potential to offer necessary crop monitoring information across well-irrigated, water stress and salinity stress in regions suffering lack of freshwater resources.

## 1. Introduction

Moisture and salinity stress are two of the most limiting factors in crop production. Both are significant factors influencing crop growth and final productivity in arid and semi-arid climates. In addition, climate change and global warming contribute to high rates of evaporation and, thus, a shortage of fresh water supplies related to drought, which is a challenge to global food security [1,2,3]. Subsequently, soil moisture and salinity stress-induced disruption in plant function leads to a significant decrease in many morpho-physiological plant characteristics, such as chlorophyll content, water status, photosynthetic performance, stomatal conductance and biomass accumulation, which ultimately causes significant losses in the quantity of grain yield [4,5,6,7]. Therefore, the timely detection of water deficiency and salinity stress effects on large scales and the application of the right amount of water at the right time is important for assuring high yields and minimizing water loses.

Estimates of crop biophysical and biochemical traits, as well as other related properties, from remotely sensed data are critical for avoiding crop yield losses [7,8]. Satellite platforms of high spatial and spectral resolution and radiometric ground-based data capabilities are suitable for estimating crop traits and many studies have demonstrated the potential of remotely sensed data in mapping vegetation [3,7,8,9]. As monitoring plant status using the sample-point technique at a regional scale is tedious, laborious, and expensive, several studies have demonstrated the potential of remotely sensed data to assess plant traits. Therefore, remote sensing can be a non-destructive technique that is simple, fast, and inexpensive to perform [9,10,11].

Abiotic stress causes significant changes in the biophysical and biochemical properties of vegetation canopies. The spectral signatures reflected from the canopy at particular wavelengths in the visible (VIS) and near infrared (NIR) regions of the spectrum change dramatically as a result of these effects [12,13,14,15,16,17].

In the developed world, airborne imagery is used to track various aspects of the Earth’s surface, especially vegetation. Some studies have shown the ability of airborne measurements to track vegetation and crop health [11,18,19]. However, obtaining airborne images is extremely expensive per unit area of ground cover. When comparing coverage areas, airborne images cover a smaller area than various satellite sensors. With the availability of free medium satellite images such as sentinel 1 and 2, the cost of monitoring agricultural crops will be far cheaper in comparison to airborne missions [6]. Another downside of airborne missions is that they are usually one-time operations, while satellite missions allow for continuous monitoring of the Earth. Using this type of imagery is difficult due to the high cost of continuous crop monitoring and the problems associated with geometric correction using airborne remote sensing in management applications [11].

Satellites with high spatial and spectral resolution have been increasingly used in precision farming in recent years. Higher-resolution satellite sensor technology (e.g., QuickBird and GeoEye1 sensors) can provide a reliable tool in site-specific management. The spatial resolution gap between satellite and airborne images has shrunk dramatically due to these types of satellite sensors [18]. The low revisit duration (1–3 days) is one of the key advantages of these satellites, as it is difficult to achieve with many other satellite systems [20]. Previous studies used QuickBird satellite images to estimate plant traits such as biochemical and biophysical parameters in crops. These include for instance estimating chlorophyll content [21], the estimation of grain yield [22,23], measuring leaf area index [24,25], the identification of nitrogen status [26], estimating above-ground biomass [10,27], the detection of crop disease [28,29], estimating crop evapotranspiration [30] and estimating dry matter [31,32].

Several satellite-based vegetation spectral indices (vegetation-SRIs) have been used to determine crop growth and final grain yield. For instance, the normalized difference vegetation index (NDVI) is often used to evaluate crop growth and yield. Leaf area index (LAI) and biomass fresh weight (BFW) were found to be responsive to the simple ratio (SR) and the green NDVI (GNDVI) [31,33]. The red edge NDVI (NDVIre) and red edge dependent vegetation indices like the red edge triangle vegetation index (RTVI) have been found to be responsive to LAI and above-ground biomass [24]. Therefore, using high-resolution satellite images such as QuickBird to measure these indices can be a reliable method for mapping within-field variability at large scales. The application of remote hyperspectral sensing, using these indices which include spectra band from the VIS and NIR, offers an easy, fast, and non-destructive approach for assessing various related plant traits compared to classical methods [15,16,17,18,19,20]. The indirect effects, which are manifested by changes in reflectance in the VIS and NIR ranges, are linked to leaf and canopy properties such as leaf pigments, leaf structure, and scattering, which change as a result of stress factors [16,17].

Weather conditions in many regions worldwide may restrict the acquisition of satellite images especially cloudy regions such as European countries which can be affected in the performance of spectral indices when estimating the crop traits [20,21,22,23,24]. Also, it is difficult sometimes to have both in situ measurements and satellite data on the same days which lead to little difference of the relationship between extracted SRIs from both platforms. However, such satellite imagery may offer farmers with better understanding the change in crop traits in other countries like Egypt since the changes in solar illumination are not that great around noon time [6].

Furthermore, in recent years, proximal remote-sensing technologies based on spectral reflectance have been used to accurately track crops during their growing period to promote good agricultural decision-making [34,35,36,37,38,39]. The advantage of proximal remote sensing is that the spectral measurements are taken close to the plant canopy to eliminate the effect of environmental conditions like the clouds and can be used as references for the satellite measurements [40,41]. In this study, the performance of vegetation-SRIs extracted from radiometric ground-based data and QuickBird satellite imagery were evaluated in full irrigation, water and salinity stress conditions.

Multivariate regression models, such as the partial least square regression (PLSR) and multiple linear regression (MLR), have substantially increased the efficiency of predicting the plant traits based on spectrum data. The PLSR and MLR have been proposed to resolve the strong multi-collinear and noisy variables in SRIs or spectral band data and to efficiently assess the measured traits [36,42,43,44]. Both methods combine a large number of SRIs or spectral bands into a single index to enhance the prediction of measured traits. Therefore, using these methods, the measured trait of maize can be simultaneously estimated through several vegetation-SRIs. PLSR can reduce a large number of calculated collinear spectral factors to a few non-correlated latent factors, avoiding over-fitting or under-fitting the data, and avoiding redundant data [45].

There is little information available to evaluate the PLSR and MLR methods based on the combination vegetation-SRIs derived from both radiometric ground-based data and QuickBird satellite imagery to predict the LAI, BFW and Chlm of maize. Therefore, the purpose of this work was to (i) assess the effects of different stress conditions on three measured traits (LAI, BFW and Chlm) of maize; (ii) evaluate the performance of classification algorithms to map different crop types within the study area; (iii) evaluate the efficiency of radiometric ground-based data and QuickBird satellite imagery based on vegetation-SRIs to assess the three measured traits of maize across full irrigation, water stress and salinity stress conditions; and (v) evaluate the performance of PLSR and MLR models based on vegetation-SRI-derived radiometric ground-based data and QuickBird satellite imagery and their combination to predict the three measured traits of maize.

## 2. Materials and Methods

### 2.1. Study Site Description

The study area is located in south-west Alexandria, Egypt (latitude of 30°55′50″ and longitude of 29°53′35.6″). To have different growing conditions for maize such as full irrigation, water and salinity stress, 15 fields were chosen (Figure 1). The weather in this region is characterized by hot summers and mild winters. The detailed climatic parameters of the study area during the experimental periods are shown in Table 1. During field visits, soil samples were collected from random fields to identify some soil chemical and physical properties. The soil at these sites is mainly a sandy loam, as these sites have been recently reclaimed from the Eastern Desert. Flood irrigation is the main irrigation system used within the study area. In this region and with the system of traditional irrigation fields at the end of an irrigation canal network, many fields suffer from water scarcity. Farmers at the end of the irrigation network are forced to use agricultural drainage water with high salinity levels as an alternative to fresh water to irrigate their crops, even with the risk of salinity stress. Other farmers wait until fresh water is available to irrigate their crops and, therefore, subject them to water stress, which leads to reductions in crop growth traits. In the 15 different fields, the maize was sown in the first week of May and was harvested in the last week of September.

### 2.2. Measured Plant Traits of Maize

Three samples from each field of study were collected immediately following reflectance measurements from maize canopies to quantify different measured traits (LAI, BFW and Chlm). Following the collection of the spectral data, a square area of 1 m length from the same locations of spectra acquisition was sampled at the ground level, and its above-ground biomass fresh weight was immediately identified. This was repeated three times and the average values were used to keep variations at a minimum. LAI was calculated as the ratio between total leaf area of a certain number of harvested plants and the area occupied by these plants. After calculating the leaf area for each sample point, the LAI was calculated by dividing the total leaf area for each sample by the area occupied by these plants according to Elmetwalli [6] as follows:LAI = LA/OA(1)
where LAI is the leaf area index, LA is the leaf area per sample, and OA is the occupied land area.

Chlorophyll was measured during field work visits using a hand-held SPAD (Soil Plant Analysis Development)-502 m (Minolta, Osaka, Japan). Apical leaves were chosen to measure Chlm. To reduce variability, Chlm was measured at three different locations on each leaf from the leaf tip to the leaf base and the average of these three readings was calculated.

### 2.3. Ground-Based Remote-Sensing Measurements

For maize, radiometric ground-based measurements were undertaken throughout 15 fields in the study area during the summer growing season of 2007, concurrent with the acquisition of QuickBird satellite imagery. From June 28 to 30, spectra were obtained from random fields, taking into account the scale of the field and the stress status of these fields. The reflectance of crop canopies was measured using an ASD FieldSpec handheld spectroradiometer. The ASD FieldSpec has two units, one connected to a diffuser to measure the light radiation as a reference signal and the second unit captures the spectral reflectance from the canopy at 300 and 1000 nm and is interpolated to a final spectral resolution of 0.5 nm. The instrument was pointed at the same angle (nadir position) to minimize changes. To keep illumination difference at a minimum, the canopy reflectance was calculated by using a calibration factor obtained from a white reference standard which was a custom built to correct readings from the spectrometer unit. This was done prior collecting reflectance at every location. Finally, the spectral reflectance was smoothed to eliminate noise at the beginning and end of the electromagnetic spectrum by passing it through a 5 nm running mean filter over the whole spectrum. Spectral reflectance from the visible (VIS) and near infrared (NIR) was used to calculate the selected spectral indices. To keep consistent illumination over field work visits, spectra measurements were restricted at noon to keep the variation at a minimum and to minimize the influence of changes in solar zenith angle. During spectra collection, an iron stand of 3 m height was used to keep the spectrometer at a consistent distance from the soil surface.

### 2.4. Remote-Sensing Image Acquisition, Processing and Analysis

QuickBird high spatial resolution images of the maize crop within the study area were acquired. The QuickBird satellite is a high spatial resolution satellite comprising four multi spectral bands (blue, green, red and near infrared) of 2.4 m spatial resolution. The QuickBird satellite image of maize fields was acquired at 09:13 h GMT on 29 June 2007. The image was radiometrically corrected by the supplier of the image (Infoterra group based in the UK). The images were geo-corrected using an image to image technique using ground control points (GCP) collected for fixed points during various field visits covering the entire study site. The images were atmospherically corrected using the FLAASH (Fast Line-of-sight Atmospheric Analysis of Hypercubes) module technique in ENVI v4.9. FLAASH is considered to be more accurate than non-physics-based models such as QUAC (QUick Atmospheric Correction) and is a remarkably established atmospheric compensation algorithm and it is mainly supportive for the majority of multi and hyper spectral remote-sensing platforms. Images were also classified using both unsupervised classification (k-means) and supervised classification (MLC; Maximum Likelihood Classification) to identify different crops in each image. K-means unsupervised and maximum likelihood supervised algorithms were used to identify different crops in the study site. K-means is one of the commonly used unsupervised classification algorithms and during the calculation of overall classification efficiency and Kappa coefficient. We tried both k-means and iso data algorithms and k-means produced higher efficiencies. These two algorithms were performed on QuickBird using the ENVI v4.9 package. The advantage of the k-means is that no additional ground reference points are required to perform the classification. Unlike the k-means technique, MLC requires a reference dataset during imagery acquisition to perform this algorithm. A confusion matrix was derived for both k-means and MLC classifications of the QuickBird satellite imagery. To perform the supervised algorithm, a validation dataset, which was independent from the training dataset, was created manually. The validation dataset included at least 2000 pixels for each class to avoid interference between various classes.

### 2.5. Calculating Vegetation Spectral Indices

Ten commonly used broad band vegetation-SRIs were derived from both radiometric ground-based data and QuickBird satellite imagery to assess the ability of remote sensing to detect maize traits. Table 2 summarizes the formulae of varying vegetation-SRIs along with references. The vegetation-SRIs were selected based on their sensitivity to changes in biomass, leaf pigmentation, leaf/tissue structure and plant water content.

### 2.6. Partial Least Squares Regression

Partial least squares regression (PLSR) is a versatile tool that can easily manage data when the number of input variables is much greater than the number of target variables, and the input variables have a lot of collinearity and noise [55,56]. In this study, to link the input variables (vegetation-SRIs listed in Table 1, which are derived from both radiometric ground-based data and QuickBird satellite imagery) to the output variables (three destructively measured traits), PLSR was combined with leave-one-out cross-validation (LOOCV). An important step in PLSR analysis is to select the optimal number of latent variables (LVs) in order to represent the calibration data without over-fitting or under-fitting. LVs parameter was calculated using the (LOOCV) according to the lowest value of the root means squared error (RMSE). Random 10-fold cross-validation was applied on the datasets to increase the robustness of the results as indicated by the software program Unscrambler X software version 10.2 (CAMO Software AS, Oslo). The performance of PLSR models based on SRIs derived from two methods and their combination were evaluated to predict the four measured traits of maize and wheat. The best model for both calibration (Cal.) and validation (Val.) was chosen depending on the lowest value of RMSE and mean absolute deviations (MAD) as well as the highest value for R^2^.

### 2.7. Multiple Linear Regression

Multiple linear regression (MLR) is a regression technique that analyzes a dependent parameter (destructively measured traits) using two or more independent parameters (SRIs are listed in Table 1, which were derived from both radiometric ground-based data and QuickBird satellite imagery). MLR attempts to model the linear relationship between the independent and the response (dependent) variable [36,37,38]. In addition, MLR was used to predict three measured traits as PLSR. The best model for both Cal. and Val. was also chosen depending on the lowest value of RMSE and mean absolute deviations (MAD) as well as the highest value for R^2^. The least squares approach was used to calculate the parameters using the regression equation, which minimizes the sum of the errors squared. The formula of MLR is:Y_i_ = β_0_ + β_1_xi_1_+ β_2_xi_2_ +………+ β_p_xi_p_ + ϵ(2)
where, for _i_ = *n* observations, Y_i_ = dependent variable, xi = explanatory variables, β_0_ = y-intercept (constant term), β_p_ = slope coefficients for each explanatory variable, ϵ = the model’s error term (also known as the residuals).

### 2.8. Statistical Analysis

This statistical analysis was performed using SPSS version 12.0 (SPSS Inc., Chicago, IL, USA). Data were checked for normality using the Anderson–Darling method with 95% significance level. Additionally, a simple linear regression of the relationship between the selected SRIs and the three plant traits was performed using SigmaPlot version 11.0 (Systat Software Inc., San Jose, CA, USA) to identify optimal SRIs based on the highest value for R^2^. The significance level of the coefficients of determination (R^2^) for these relationships was set at the 0.05 probability level.

## 3. Results and Discussion

### 3.1. Impact of the Full Irrigated and Different Stress Conditions on Three Measured Traits of Maize

Fifteen fields in this study, which were exposed to full irrigation and different stress conditions such as water and salinity stress, were chosen to present the variation in the values of LAI, BFW and Chlm of maize. There were clear differences in the values of the three measured traits of maize between the different treatments. For example, LAI varied from 2.38 to 3.13 for irrigation, 1.29 to 2.17 for water stress, and 0.27 to 0.99 for salinity stress (Table 3). The BFW varied from 2.45 to 2.80 kg m^−2^ for irrigation, 1.20 to 1.83 kg m^−2^ for water stress, and 0.25 to 0.86 kg m^−2^ for salinity stress. The Chlm varied from 47.9 to 49.3 for irrigation, 42.8 to 50.1 for water stress, and 31.1 to 34.8 for salinity stress (Table 3).

Water stress presented the second order in terms of affecting measured traits after salinity stress treatments. The remarkable differences in LAI and biomass between water stress and full irrigation may be related to the photosynthetic processes that are closely linked to the growth of leaves and whole biomass. The efficiency of radiation transfer to dry matter by photosynthesis is a major factor in total above-ground biomass accumulation. Other studies agreed with our finding since they reported that water shortage during the vegetative growth stage of maize plants could inhibit the growth and leaf area of maize [57,58,59,60,61]. Insufficient available soil water weakens maize metabolism, reduces biomass accumulation, and reduces photosynthetic rate by lowering chlorophyll content in leaves, ultimately lowering maize yield [62,63,64,65,66,67,68,69].

The salinity stress strongly affected the measured traits of maize compared to the other treatments. Salinity reduces BFW and LAI by inhibiting leaf initiation and expansion, as well as internode development, and by accelerating leaf abscission [4,5]. Furthermore, under salt stress, a decrease in photosynthetic pigments such as chlorophylls a and b, as well as carotenoids, is related to a decrease in net photosynthesis rate in maize [5,70]. These findings emphasize the importance of simultaneous and frequent assessment of measured traits for improving the tolerance of maize to different stress conditions. However, improving maize tolerance requires methods that allow for simple, fast, and non-destructive evaluation of large-scale fields on a regular basis; these methods will be discussed in detail below.

### 3.2. Identifying Different Crops in the Study Area

Both classification algorithms mainly produced five different classes including maize, tomato, melon, water, and bare soil. A confusion matrix was derived for both algorithms to evaluate their ability to show the overall accuracy, accuracy for classifying each class, Kappa coefficient as a percentage, user’s accuracy, and producer’s accuracy as a percentage. The results showed that the overall classification accuracy of the MLC algorithm for different classes is high, ranging from 89.9% (for maize) to 96.8% (for water). Similar to the overall accuracy, both user’s and producer’s accuracy were also high in MLC (>0.86). It can also be seen that the classification accuracy for identifying maize crops using MLC was very high (>0.80) with high classification accuracies for other classes (Table 4).

Although the k-means classifier algorithm produced very high classification accuracy for maize (93.46) in Table 5, it showed less accuracy in identifying tomato crops, which may lead to many misclassified pixels and thus low overall accuracy. Consequently, the unsupervised classification algorithm produced very high accuracy for identifying maize fields, but the overall accuracy and classification accuracy for single classes were lower compared with the MLC algorithms. Moreover, kappa coefficient values in the MLC algorithm were comparable to those obtained from the k-means classifier. The MLC algorithm showed high overall accuracy and single classification accuracy at the same time as identifying various classes in summer.

### 3.3. Evaluation of Spectral Reflectance Indices (SRIs) Extracted from Radiometric Ground-Based Data and High Resolution Satellite Imagery to Assess Plant Traits of Maize under Different Stress Conditions

In this study, the performance of the selected vegetation-SRIs from both techniques of radiometric ground-based data and high resolution satellite images were evaluated to assess the three measured traits across well-irrigated, water stress and salinity stress conditions. Vegetation-SRIs, such as those listed in Table 2, are useful for monitoring changes in pigment content, photosynthetic efficiency, biomass accumulation, leaf area index and other aspects of vegetation canopy growth and health [39,40,41,71,72]. The results showed that the majority of SRIs extracted from radiometric ground-based data and high-resolution QuickBird satellite imagery were more effective in estimating LAI, BFW, and Chlm. The R^2^ of the significant relationships between vegetation-SRIs of radiometric ground-based data and the three measured traits varied from 0.64 to 0.89 (Figure 2). As well as with the high resolution QuickBird satellite imagery, R^2^ varied from 0.34 to 0.84 (Figure 2). For radiometric ground-based data, the relationships between GNDVI and LAI and BFW showed the highest R^2^, which was 0.98 and 0.86, respectively (Figure 2). With high resolution QuickBird satellite imagery, the relationships between GNDVI and LAI and BFW showed the highest R^2^, which was 0.80 and 0.84, respectively (Figure 2). The VI index derived from radiometric ground-based data failed to estimate the three measured traits, but the VI index derived from high-resolution QuickBird satellite imagery presented strong relationships with LAI and BFW, and moderate relationships with Chlm. In agreement with our results, many ground-based remote-sensing and satellite-based vegetation indices perform well in assessing crop growth performance and chlorophyll meter. The simple ratio (SR) and the GNDVI were noticed to be sensitive to greater LAI and BFW [31,33,73]. For example, Mzid et al. [73] found that LAI, NDVI, optimized soil adjusted vegetation index (OSAVI), and enhanced vegetation index (EVI) regressions of wheat tended to have high correlations with R^2^ values equal to or higher than 0.70. Shanahan et al. [74] found that GNDVI had higher correlations with chlorophyll meter than NDVI, with overall correlations of around 0.90 at the V15 growth stage.

Several vegetation-SRIs in this study were calculated based on the NIR and red (R) bands. In this trend, previous studies have shown that NIR and R band vegetation indices are effective for estimating the BFW and biomass dry weight (BDW) [75,76,77]. In general, the majority of vegetation-SRIs of radiometric ground-based data showed higher R^2^ with measured traits compared to the vegetation-SRIs extracted from high-resolution QuickBird satellite imagery (Figure 2). Overall, the ground-based vegetation indices and the satellite-based indices were found to be in good agreement with the measured traits. The results obtained from this study showed that radiometric ground-based measurements and high spectral resolution remote-sensing imagery have the potential to offer necessary crop monitoring information across well-irrigated, water stress and salinity a stress. Combining remote sensing and field data may be essential to optimize the plant response under different stress conditions and boost agricultural production.

### 3.4. Performance of Partial Least Square Regression (PLSR) and Multiple Linear Regression (MLR) Models to Predict the Measured Traits of Maize

Although SRIs are simple to calculate and several indices have been effective in estimating measured traits, they are constrained by their use of just a few bands and are affected by vegetation saturation or varying degrees, as well as timeliness and regional specificity [38,39,40,41,78,79]. Multivariate regression models such as PLSR and MIR have previously been shown to be alternative methods to SRIs for explaining the relationships between plant-measured traits and spectral reflectance and perform equally well or better than SRIs for estimating the changes in these traits [42,45]. In this study, the vegetation-SRIs of radiometric ground-based measurements and the same vegetation-SRIs extracted from QuickBird satellite imagery, and their combination were applied to PLSR and MLR in order to predict the LAI, BFW and Chlm. To the best of our knowledge, data fusion from different platform sensors adopted in the measured traits models and prediction of field trials has not been used.

To reduce the strongly collinear independent variables to a small minority of orthogonal factors, multivariate statistical techniques, PLSR models, based on the selected spectral indices, were tested to measured traits of maize. PLSR techniques can be used to identify optimized models that enhance the efficiency when searching for optimized relationships. The number of latent variables is used. The calibration (Cal.) models of PLSR and MLR showed the highest performance in predicting the measured traits based on the combination of vegetation-SRIs from two methods with R^2^ = 0.92–0.93 for LAI, R^2^ = 0.92–0.94 for BFW, and R^2^ = 0.79–0.87 for SPA (Table 6 and Table 7). Furthermore, validation (Val.) models showed the highest performance in predicting the measured traits based on the combination of vegetation-SRIs from two methods with R^2^ = 0.91 for LAI, R^2^ = 0.91–0.93 for BFW, and R^2^ = 0.82 for SPAD (Table 6 and Table 7). The two models of PLSR and MLR presented approximately the same performance in predicting the three measurement traits and no clear difference was found through the two methods and their combinations (Table 6 and Table 7; Appendix A and Figure 3, Figure 4 and Figure 5). The Cal. and Val. of PLSR models based on vegetation-SRIs of radiometric ground-based measurements showed a greater performance in predicting LAI, BFW and Chlm than the PLSR models of QuickBird satellite imagery (Table 6; Figure 3, Figure 4 and Figure 5).

For example, the calibration (Cal.) models showed the highest performance in predicting the measured traits based on vegetation-SRIs from radiometric ground-based measurements, with R^2^ = 0.89, root means squared error for calibration (RMSEc) = 0.32, and mean absolute deviations (MADc) = 0.28 for LAI, R^2^ = 0.91, RMSEc = 0.29, and MADc = 0.24 for BFW, and R^2^ = 0.85, RMSEc = 3.07, and MADc = 2.29 for Chlm (Table 6; Figure 3, Figure 4 and Figure 5). Ge et al. [35] found that PLSR based on spectral reflectance gave higher prediction accuracy for leaf chlorophyll content, specific leaf area and leaf water content of maize compared to vegetation indices. Hansen and Schjoerring [42] obtained a better estimate of green biomass and leaf nitrogen concentration of winter wheat in a field experiment when using PLSR rather than the best of narrow-band SRIs. In addition, Sharabian et al. [43] found that PLSR models could improve the assessment of the GY dataset (R^2^ = 0.87, RMSE = 301) and chlorophyll meter (R^2^ = 0.84, RMSE 1.94) since strong relationships exist between the predicted and measured values for a validation.

The Cal. and Val. of PLSR and MLR models based on vegetation-SRIs of radiometric ground-based measurements showed a higher performance in predicting LAI, BFW and Chlm compared to the PLSR and MLR models of QuickBird satellite imagery (Table 6 and Table 7; Appendix A and Figure 3, Figure 4 and Figure 5). For example, the Cal. models showed the highest performance in predicting the measured traits based on vegetation-SRIs from radiometric ground-based measurements, with R^2^ = 0.87, RMSEc = 0.35, and MADc = 0.30 for LAI, R^2^ = 0.89, RMSEc = 0.30, and MADc = 0.25 for BFW, and R^2^ = 0.82, RMSEc = 3.37, MADc = 2.47 for Chlm (Table 7). Han et al. [78] found that MLR based on remote-sensing spectral data produced acceptable accuracy in predicting above-ground biomass. In addition, Luo et al. [79] found that both the MLR and support vector machine (SVR) models based on spectral data showed the accuracy estimation of LAI and biomass of maize at the flowering stage since the growth and development of a maize canopy reaches a plateau at flowering, and LAI and biomass no longer grow as quickly. The overall results indicate that the proposed PLSR and MLR models based on the combination of vegetation-SRIs from two methods can improve the accuracy of plant trait estimates.

## 4. Conclusions

In this study, the performance of vegetation-SRIs extracted from radiometric ground-based data and high-resolution QuickBird satellite imagery to assess the leaf area index (LAI), biomass fresh weight (BFW) and chlorophyll meter (Chlm) of maize across full irrigation, water stress and salinity stress in the Nile Delta of Egypt was investigated. The performance of partial least squares regression (PLSR) and multiple linear regression (MLR) were evaluated to estimate the three measured traits based on vegetation-SRIs extracted from two methods and their combination. Our results indicated that the majority of vegetation-SRIs extracted from both methods could assess plant traits. In general, the vegetation-SRIs of radiometric ground-based data showed higher R^2^ with measured traits compared to the vegetation-SRIs extracted from high resolution satellite imagery. Overall, the ground-based vegetation-SRIs and the satellite-based indices were found to be in good agreement to assess the measured traits. The calibration (Cal.) and validation (Val.) of PLSR and MLR models based on vegetation-SRIs of radiometric ground-based data showed a higher performance in predicting LAI, BFW and Chlm compared to the PLSR and MLR models of QuickBird satellite imagery. Both Cal. and Val. models of PLSR and MLR showed the highest performance in predicting the three measured traits based on the combination of vegetation-SRIs from radiometric ground-based data and high resolution QuickBird satellite imagery. Finally, this work’s main conclusion is that both methods of ground-based remote sensing and high-resolution QuickBird satellite imagery can provide a useful tool for estimating maize crop traits. Our models are simple and can be used in non-destructive and large-scale methods to assess plant morpho-physiological traits quickly and accurately.

## Figures and Tables

**Figure 1 sensors-21-03915-f001:**
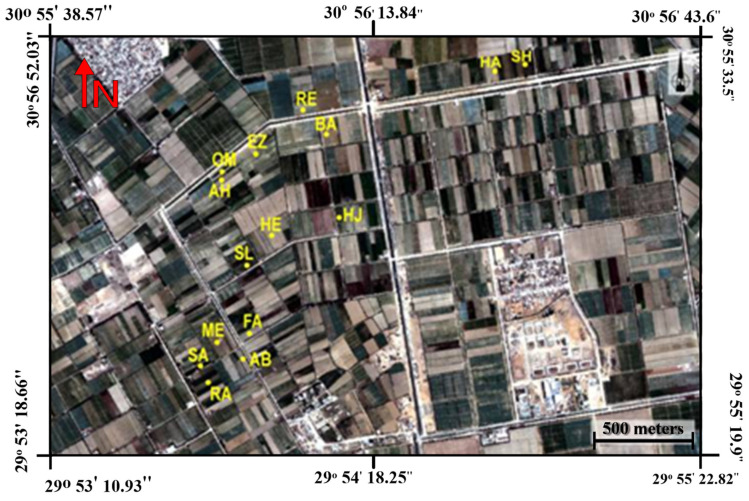
QuickBird satellite image showing different fields within the study area south-west Alexandria, Egypt. SH, HA, RE, BA, EZ, OM, AH, HJ, HE, SL, FA, ME, SA, FA and RA are the codes of 15 fields.

**Figure 2 sensors-21-03915-f002:**
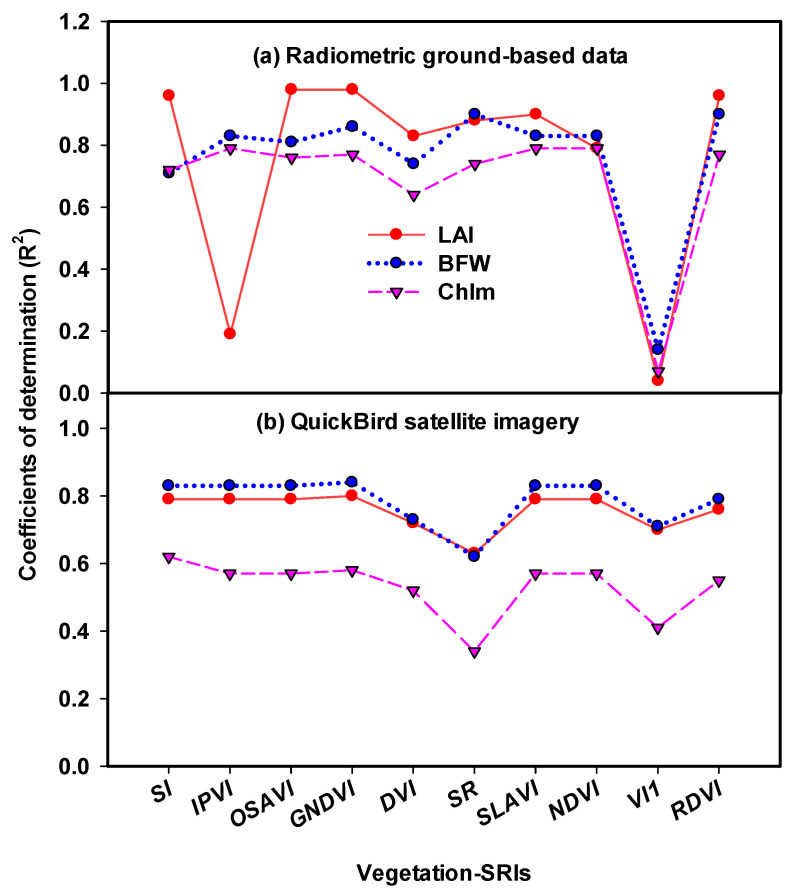
Coefficients of determination (R^2^) for the relationship between selected spectral reflectance indices (SRIs) in Table 1 extracted from radiometric ground-based data and QuickBird satellite imagery with leaf area index (LAI), biomass fresh weight (BFW) and chlorophyll meter (Chlm) of maize under full irrigated and different stress conditions.

**Figure 3 sensors-21-03915-f003:**
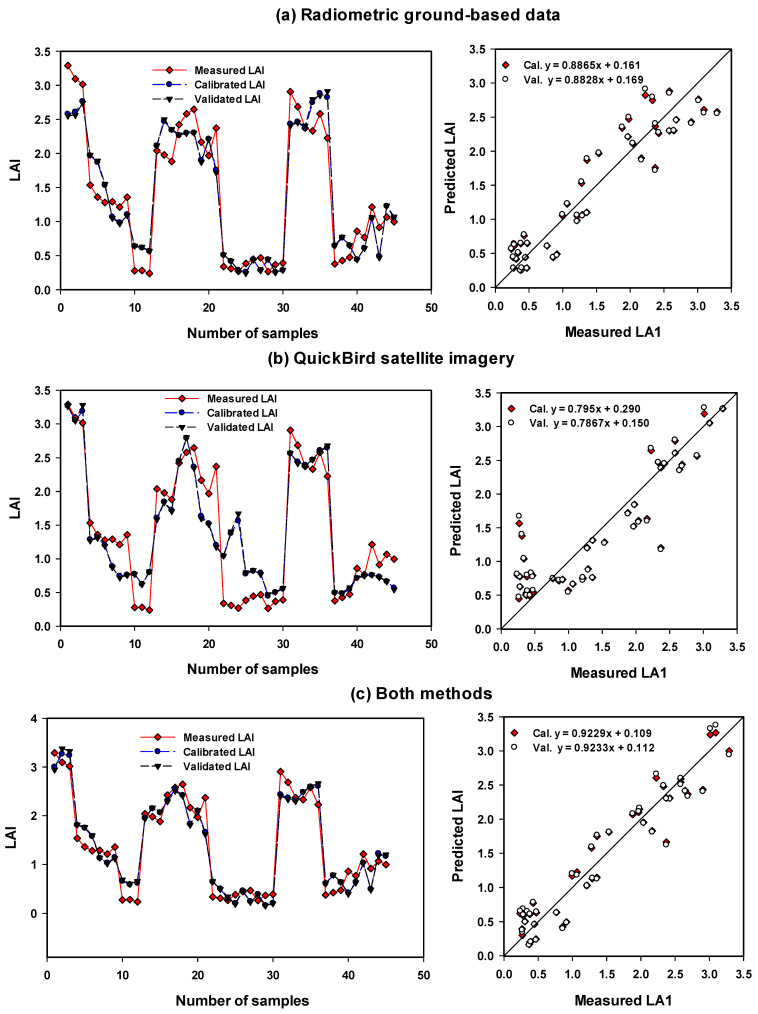
Comparison between measuring series, calibrating series and validating series for leaf area index (LAI) of maize using the PLSR models based on selected vegetation-(SRIs) extracted from (**a**) radiometric ground-based data, (**b**) extracted from QuickBird satellite imagery and, (**c**) extracted from both methods.

**Figure 4 sensors-21-03915-f004:**
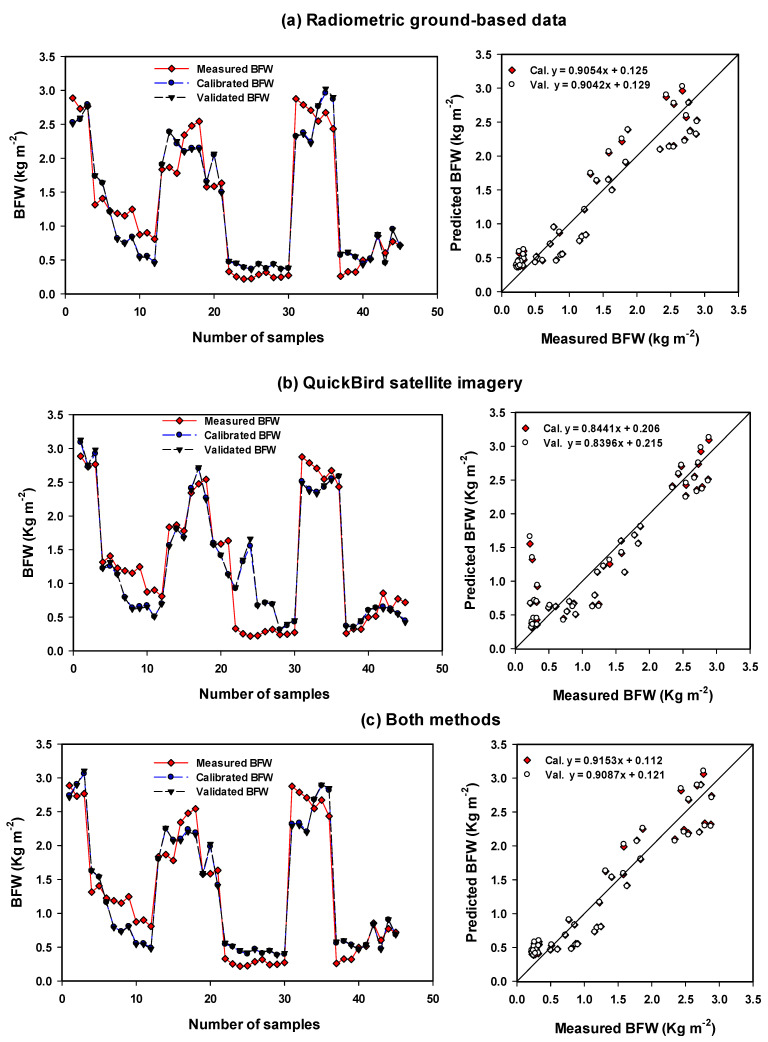
Comparison between measuring series, calibrating series and validating series for biomass fresh weight (BFW) of maize using the PLSR models based on selected vegetation-(SRIs) extracted from (**a**) radiometric ground-based data, (**b**) extracted from QuickBird satellite imagery, and (**c**) extracted from both methods.

**Figure 5 sensors-21-03915-f005:**
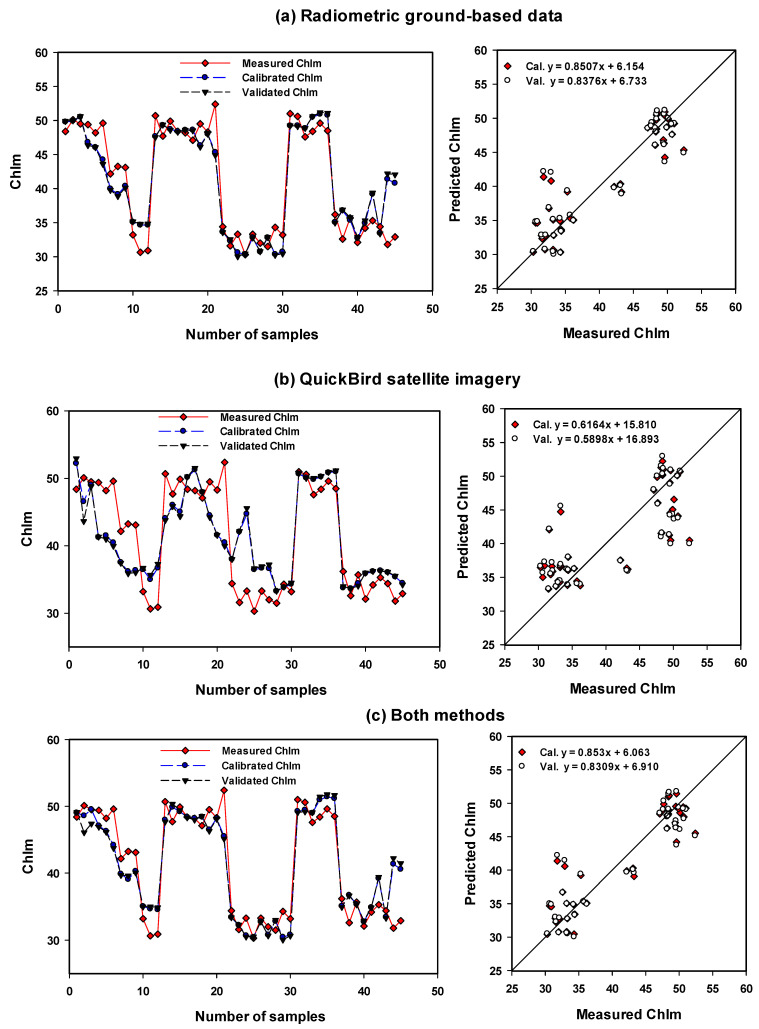
Comparison between measuring series, calibrating series and validating series for chlorophyll meter (Chlm) of maize using the PLSR models based on selected vegetation-(SRIs) extracted from (**a**) radiometric ground-based data, (**b**) extracted from QuickBird satellite imagery, and (**c**) extracted from both methods.

**Table 1 sensors-21-03915-t001:** Agro-climatological data for the study area recorded at Borg Alarab Station (latitude of 31.2° and longitude of 29.916°). This data is just from the period of our experimentation.

Year	Month	T_max_ °C	T_min_ °C	U_2_ (km/Day)	RH (%)	SH (h)
2007	April	24.2	13.5	1.65	83.4	9.3
	May	26.9	16.9	1.41	82.1	10.1
	June	29.4	20.6	1.39	84.6	10.7
	July	29.9	23.1	1.50	86.1	10.3
	August	30.7	23.4	1.31	79.2	9.9

T_max_, T_min_, U_2_, RH, and SH indicate maximum temperature, minimum temperature, average wind speed, relative humidity, and sunshine hours.

**Table 2 sensors-21-03915-t002:** Formulae of selected spectral reflectance indices (SRIs) and references collected from the literature.

Vegetation-SRIs	Formulae	Reference
Stress index (SI)	Red/NIR	[46]
Infra-red percentage vegetation index (IPVI)	NIR/(NIR+Red)	[47]
Optimized soil adjusted vegetation index (OSAVI)	((NIR − Red)/(NIR + Red + L)) × (1 + L), L = 0.16	[48]
Green normalized difference vegetation index (GNDVI)	(NIR − green)/(NIR + green)	[18]
Difference vegetation index (DVI)	NIR − Red	[49]
Simple ratio (SR)	NIR/Red	[50]
Specific leaf area vegetation index (SLAVI)	NIR/(Red + NIR)	[51]
Normalized difference vegetation index (NDVI)	(NIR − Red)/(NIR + Red)	[52]
Vegetation index (VI)	NIR/(green-1)	[53]
Renormalized difference vegetation index (RDVI)	$\sqrt{NDVI\;x\;VI}$	[54]

**Table 3 sensors-21-03915-t003:** Mean and standard deviation (SD) of leaf area index (LAI), biomass fresh weight (BFW) and chlorophyll meter (Chlm) of maize.

Fields	Field Code	Treatment	LAI	SD	BFW (kg m^−2^)	SD (kg m^−2^)	Chlm	SD
Field 1	SA	Irrigated	3.13 a	0.14	2.80 a	0.08	49.3 a	0.86
Field 2	FA	Water stress	1.39 f	0.13	1.32 e	0.10	49.1 a	0.76
Field 3	SL	Water stress	1.29 f	0.08	1.20 e	0.05	42.8 b	0.59
Field 4	FA	Salinity stress	0.27 h	0.02	0.86 f	0.05	31.6 d	1.41
Field 5	HE	Water stress	1.97 e	0.08	1.83 c	0.05	49.4 a	1.55
Field 6	RA	Irrigated	2.55 bc	0.12	2.45 b	0.10	47.9 a	0.70
Field 7	HJ	Water stress	2.17 de	0.20	1.60 d	0.03	50.1 a	2.11
Field 8	EZ	Salinity stress	0.31 h	0.04	0.27 h	0.06	33.1 cd	1.41
Field 9	AH	Salinity stress	0.43 h	0.05	0.28 h	0.05	31.9 d	1.50
Field 10	OM	Salinity stress	0.34 h	0.07	0.25 h	0.02	33.0 cd	1.41
Field 11	ME	Irrigated	2.67 b	0.27	2.79 a	0.09	49.0 a	1.86
Field 12	AB	Irrigated	2.38 cd	0.18	2.55 b	0.12	48.8 a	0.67
Field 13	EZ	Salinity stress	0.43 h	0.05	0.30 h	0.04	34.8 c	1.95
Field 14	RE	Salinity stress	0.95 g	0.23	0.62 g	0.21	33.9 cd	1.63
Field 15	BA	Salinity stress	0.99 g	0.08	0.70 g	0.09	33.0 cd	1.31

The mean values with the same letter are not statistically different (*p* > 0.05) between different treatments among different fields.

**Table 4 sensors-21-03915-t004:** Confusion matrix results for the MLC (Maximum Likelihood Classification) algorithm of maize and other crops in south-west Alexandria, Egypt.

Class	Ground Truth (%)						User’s Accuracy
	Maize	Tomato	Melon	Bare Soil	Water	Total	
Unclassified	0.00	0.00	0.00	0.00	0.00	0.00	(%)
Maize	89.87	4.66	5.1	4.07	0.00	20.81	86.70
Tomato	3.28	93.67	1.67	0.19	0.00	19.98	94.82
Melon	2.06	0.19	92.40	0.00	0.09	19.25	97.55
Bare soil	4.78	1.49	0.46	95.74	3.12	20.60	90.40
Water	0.00	0.00	0.37	0.00	96.79	19.36	99.61
Total	100	100	100	100	100	100.00	
Producer’s accuracy (%)	89.87	93.67	92.4	95.74	96.79		
Kappa coefficient	0.921						
Overall accuracy	93.67%						

**Table 5 sensors-21-03915-t005:** Confusion matrix results for k-means algorithm of maize and other crops in south-west Alexandria, Egypt.

Class	Ground Truth (%)						User’s Accuracy
	Maize	Tomato	Melon	Bare Soil	Water	Total	
Unclassified	0	0	0	0	0	0	(%)
Maize	93.46	41.88	3.6	0.67	1.67	26.56	61.91
Tomato	6.09	58.12	2.43	0.96	32.32	20.63	59.90
Melon	0.23	0.00	93.77	3.94	0.00	19.97	95.73
Bare soil	0.23	0.00	0.00	66.01	0.29	13.39	99.80
Water	0.00	0.00	0.19	0.00	94.13	19.46	99.26
Total	100	100	100	100	100	100	
Producer’s accuracy (%)	93.46	58.12	93.77	94.13	66.01		
Kappa coefficient	0.758						
Overall accuracy	80.62%						

**Table 6 sensors-21-03915-t006:** Results of calibration (R^2^_cal_, RMSE_C_ and MADc), and 10-fold cross-validation (R^2^_cal_, RMSEv and MADv) partial least square regression (PLSR) models of the relationship between selected spectral reflectance indices extracted in Table 1 from radiometric ground-based data, QuickBird satellite and both of them with leaf area index (LAI), biomass fresh weight (BFW) and chlorophyll meter (Chlm) of maize.

SRIs Types	Measured Variables	LVs	Calibration			Validation		
			R^2^_cal_	RMSEc	MADc	R^2^_val_	RMSEv	MADv
Radiometric ground-based data	LAI	2	0.89 ***	0.32	0.28	0.88 ***	0.34	0.29
	BFW	1	0.91 ***	0.29	0.24	0.90 ***	0.30	0.25
	Chlm	2	0.85 ***	3.07	2.29	0.85 ***	3.33	2.47
QuickBird satellite	LAI	2	0.79 ***	0.43	0.32	0.77 ***	0.45	0.34
	BFW	2	0.84 ***	0.37	0.26	0.83 ***	0.39	0.28
	Chlm	2	0.62 ***	4.92	3.92	0.57 ***	5.23	4.19
Both methods	LAI	3	0.92 ***	0.27	0.23	0.91 ***	0.29	0.25
	BFW	1	0.92 ***	0.27	0.23	0.91 ***	0.29	0.25
	Chlm	3	0.86 ***	3.05	2.26	0.82 ***	3.40	2.55

Levels of significance: ***: *p* < 0.001, RMSEc is root means squared error for calibration, MADc is mean absolute deviations for calibration, RMSEv is root means squared error for validation, and MADv is mean absolute deviations for validation.

**Table 7 sensors-21-03915-t007:** Results of calibration (R^2^_cal_, RMSE_C_ and MADc), and 10-fold cross-validation (R^2^_cal_, RMSEv and MADv) of multiple linear regression models (MLR) of the relationship between selected spectral reflectance indices extracted in Table 1 from radiometric ground-based data, QuickBird satellite and both of them with leaf area index (LAI), biomass fresh weight (BFW) and chlorophyll meter (Chlm) of maize.

SRIs Types	Measured Variables	Calibration			Validation		
		R^2^_cal_	RMSE_C_	MADc	R^2^_val_	RMSE_V_	MADv
Radiometric ground-based data	LAI	0.89 ***	0.34	0.28	0.87 ***	0.35	0.30
	BFW	0.91 ***	0.29	0.24	0.89 ***	0.30	0.25
	Chlm	0.85 ***	3.18	2.29	0.82 ***	3.37	2.47
QuickBird satellite	LAI	0.80 ***	0.45	0.32	0.77 ***	0.46	0.34
	BFW	0.84 ***	0.38	0.27	0.83 ***	0.39	0.28
	Chlm	0.62 ***	5.09	3.91	0.57 ***	5.26	4.17
Both methods	LAI	0.93 ***	0.28	0.22	0.91 ***	0.29	0.24
	BFW	0.94 ***	0.24	0.19	0.93 ***	0.26	0.21
	Chlm	0.85 ***	3.18	2.31	0.82 ***	3.41	2.61

Levels of significance: ***: *p* < 0.001.

## Data Availability

Data are contained within the article and in the Appendix A.

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
