# Peer review of "Integration of Radiometric Ground-Based Data and High-Resolution QuickBird Imagery with Multivariate Modeling to Estimate Maize Traits in the Nile Delta of Egypt"

_sensors, 2021, doi:10.3390/s21113915_

Round 1

Reviewer 1 Report

Line 61-66 Please clarify which reference refers to what sentence.

Line 68: Change “was” to “IS” as this activity is ongoing.

Line 71-73: Satellite data is also expensive – are these sentences inferring the utility of free data from space such as Landsat? Please clarify.

Line 75-82: Still no comparison of airborne vs satellite costs.

Line 84-86: More explanation on the utility of these indices plus their limitations is needed. How does shadowing and atmospheric conditions effect the results?

Table 1: Just to clarify - is this data from just the period of your experimentation or an average over multiple years?

Line 136: Please reference this equation.

Line 158: Why was 300 nm used and not 400 nm? Was UV analysis part of this work?

Line 159: Please provide a bit more detail on calibration method. What white standard was used? Was it integrated with the instrument or custom built?

Line 161: Please describe “magnetic spectrum”

Line 162: How was different illumination taken into account? Was the spectrometer consistently pointed at the same angle from the sun? How was shadowing and specular reflections kept consistent?

Line 172-179: References are needed for this section – why was the FLASH atmospheric correction model chosen over others? Also need a bit more explanation as to why k-means was the best choice here.

Line 207-222: need references here.

Line 253: Appears to be a repetition of Materials and Methods

Line 282-291: Appears to be a mix of literature review and discussion. Please rework to ensure previous research (eg Lines 282-285) are in the appropriate section. Line 290-291 would make a better introduction to this section.

Line 295-296: Low to moderate R2 values – what relationship can actually be inferred from the lower values?

Line 301-311: The messaging is getting lost as this section seems to be a mix of literature review and findings. Please clarify and separate so you can demonstrate a clear message on how your results support the utility of satellite-based remote sensing.

Line 316-318: Do you have a figure/table to illustrate this? If so please reference it here.

Author Response

  1. May 2021

Dr. Vittorio M.N. Passaro 

Editor-in-Chief

Sensors

Dear Prof. Passaro,

Please find attached the revised manuscript titled ‘Integration of radiometric ground-based data and high resolution QuickBird imagery with Multivariate modeling to estimate maize traits in the Nile Delta of Egypt’. Manuscript ID: sensors-1209459, authored by Adel El-Metwalli, Andrew N. Tyler, Farahat S. Moghanm, Saad A.M. Alamri, Ebrahem M. Eid, and Salah Elsayed.

On behalf of my co-authors, I thank you very much for giving us the opportunity to revise our manuscript. We appreciate the positive and constructive comments and suggestions provided by the reviewers on our manuscript. We have carefully studied the reviewers’ comments and have made revisions that are indicated using track changes in the revised version of the manuscript. We have tried our best to revise our manuscript according to the reviewers’ comments. Please find attached the revised version of our manuscript, which we would like to submit for your kind consideration. Once again, we would like to express our great appreciation to you and the reviewers for the comments on our manuscript.Please find below our detailed responses to each of the points raised.

-----------------------------------------------------------------------------------------------------------------Line 61-66 Please clarify which reference refers to what sentence.

Response: Many thanks for this comment. The reference for each sentence was added in right place and the sentences have been editing.

-----------------------------------------------------------------------------------------------------------------

Line 68: Change “was” to “IS” as this activity is on-going.

Response: It was modified from was to is.

-----------------------------------------------------------------------------------------------------------------

Line 71-73: Satellite data is also expensive – are these sentences inferring the utility of free data from space such as Landsat? Please clarify.

Response: Thanks for your comment but here we are talking about reasonable resolution satellite imagery which is still cheaper than airborne imagery. When considering the coverage area of each type (airborne and satellite), the cost of airborne will be far expensive in addition to a great number of flights to cover a certain area which will take too much time in comparison to satellite sensors. So even with paid satellite imagery it is still much cheaper. Moreover, many satellite sensors with medium resolution can be obtained with no charge.

-----------------------------------------------------------------------------------------------------------------  

Line 75-82: Still no comparison of airborne vs satellite costs.

Response:  As we explained in the previous point and added in the revised manuscript (line 74-76) using satellite images either free or paid images it is still comparable in terms of cost.

-----------------------------------------------------------------------------------------------------------------

Line 84-86: More explanation on the utility of these indices plus their limitations is needed. How does shadowing and atmospheric conditions affect the results?

Response: The utility of these indices plus their limitations as well as shadowing and atmospheric conditions affect the results was written in section of introduction.

-----------------------------------------------------------------------------------------------------------------

Table 1: Just to clarify - is this data from just the period of your experimentation or an average over multiple years?

Response: This data is just from the period of our experimentation and the title 1 of the Table was modified.

-----------------------------------------------------------------------------------------------------------------

Line 136: Please reference this equation.

Response: The reference this equation was added.

-----------------------------------------------------------------------------------------------------------------

Line 158: Why was 300 nm used and not 400 nm? Was UV analysis part of this work?

Response: The spectral range of the ASD field used in this research was 300-1075 nm so we processed the entire obtained range.  Only just the visible (VIS) and near infrared (NIR) portions of the electromagnetic spectrum was used to calculated spectral indices. So, UV was not a part of our work. The section of 2.3 ground-based remote sensing measurements was modified according to reviewer comments to be clear for the reader.

-----------------------------------------------------------------------------------------------------------------

Line 159: Please provide a bit more detail on calibration method. What white standard was used? Was it integrated with the instrument or custom built?

Response: Sure, prior to the acquisition of reflectance from plant canopies, the spectroradiometer was calibrated to a white reference panel which was a custom built. Once the white reference taken, different scans were collected. The section of 2.3 ground-based remote sensing measurements was modified according to reviewer comments to be clear for the reader.

-----------------------------------------------------------------------------------------------------------------

Line 161: Please describe “magnetic spectrum”

Response: Thanks for your comment. It should be electromagnetic and has been changed in the revised manuscript. Electromagnetic spectrum refers to different regions including blue, green, red and near infrared which are captured by the spectroradiometer.

-----------------------------------------------------------------------------------------------------------------  

Line 162: How was different illumination taken into account? Was the spectrometer consistently pointed at the same angle from the sun? How was shadowing and specular reflections kept consistent?

Response: To keep consistent illumination over field work visits, spectra measurements were restricted noon time to keep the variation at a minimum. The iron stand was used to point the instrument at nadir position. In addition, the changes in solar illumination in the study site are not that much around noon time. The section of 2.3 ground-based remote sensing measurements was modified according to reviewer comments to be clear for the reader.

-----------------------------------------------------------------------------------------------------------------

Line 172-179: References are needed for this section – why was the FLASH atmospheric correction model chosen over others? Also need a bit more explanation as to why k-means was the best choice here.

Response: FLASH is considered to be more accurate than non-physics based models such as QUAC and is a remarkably established atmospheric compensation algorithm and it is mainly supportive for the majority of multi and hyper spectral remote sensing platforms. FLASH module has been used by many studies (Lopez-Serrano et al., 2016) showing better performance this is why we used it in our research. K-means is one of the commonly used unsupervised classification algorithms and during the calculation of overall classification efficiency and Kappa coefficient we tried both k-means and iso data algorithms and k-means produced higher efficiencies.  

-----------------------------------------------------------------------------------------------------------------

Line 207-222: need references here.

Response: The references were added.

-----------------------------------------------------------------------------------------------------------------

Line 253: Appears to be a repetition of Materials and Methods

Response: This sentence has been deleted.

-----------------------------------------------------------------------------------------------------------------  

Line 282-291: Appears to be a mix of literature review and discussion. Please rework to ensure previous research (eg Lines 282-285) are in the appropriate section. Line 290-291 would make a better introduction to this section.

Response: The first part of this paragraph has been moved to the introduction section (Line 72-77). As well as line 290-291 was used as better introduction to this section.

-----------------------------------------------------------------------------------------------------------------

Line 295-296: Low to moderate R2 values – what relationship can actually be inferred from the lower values?

Response: The lower values of R2 basically refer to week relationships between spectral reflectance indices extracted from either satellite or ground based radiometric measures which can be related to stress or to the interference from soil background when plant biomass does not cover the entire ground surface.

-----------------------------------------------------------------------------------------------------------------    

Line 301-311: The messaging is getting lost as this section seems to be a mix of literature review and findings. Please clarify and separate so you can demonstrate a clear message on how your results support the utility of satellite-based remote sensing.

Response: The section (3.3 Evaluation of SRIs extracted from radiometric ground-based data and high-resolution satellite imagery to assess plant traits of maize under different stress conditions) was modified and all unnecessary pervious works was removed.

-----------------------------------------------------------------------------------------------------------------

Line 316-318: Do you have a figure/table to illustrate this? If so please reference it here.

Response: It is shown in Figure 2. You can see higher R2 for the relationship between SRI extracted from radiometric ground – based indices than those obtained from satellite – based indices.

-----------------------------------------------------------------------------------------------------------------

I hope the explanation given above adequately addresses all reviewers’ comments. I would appreciate if the revised version of our manuscript would be considered for publication in Sensors.

Yours

Ebrahem M. Eid

[Kafrelsheikh University]

[Botany Department, Faculty of Science, Kafrelsheikh University, Kafr El-Sheikh 33516, Egypt]

[Phone number: 002010 22648840]

[Email address: ebrahem.eid@sci.kfs.edu.eg]

Reviewer 2 Report

This manuscript investigates in detail the maize traits using in-suit observations and QuickBird images in the Nile Delta of Egypt. It's informative and interesting. I have some minor suggestions for improvement.

1. The abstract is not clear and needs to be rewritten.

2. The traits and characteristics of plants have two different meanings. This manuscript is actually a remote sensing inversion of some maize indices. 

3. "3. Results and Discussion" is followed by "5. Conclusions"? I suggest writing results and discussion separately for more clarity.

4. Two methods are used to classify maize. I think the difference between the selected species and maize is very big, and the plants with less difference from maize in the area should be selected.

5. What do the letters a, b, c, and d after the numbers in Table 3 mean?

Author Response

  1. May 2021

Dr. Vittorio M.N. Passaro 

Editor-in-Chief

Sensors

Dear Prof. Passaro,

Please find attached the revised manuscript titled ‘Integration of radiometric ground-based data and high resolution QuickBird imagery with Multivariate modeling to estimate maize traits in the Nile Delta of Egypt’. Manuscript ID: sensors-1209459, authored by Adel El-Metwalli, Andrew N. Tyler, Farahat S. Moghanm, Saad A.M. Alamri, Ebrahem M. Eid, and Salah Elsayed.

On behalf of my co-authors, I thank you very much for giving us the opportunity to revise our manuscript. We appreciate the positive and constructive comments and suggestions provided by the reviewers on our manuscript. We have carefully studied the reviewers’ comments and have made revisions that are indicated using track changes in the revised version of the manuscript. We have tried our best to revise our manuscript according to the reviewers’ comments. Please find attached the revised version of our manuscript, which we would like to submit for your kind consideration. Once again, we would like to express our great appreciation to you and the reviewers for the comments on our manuscript.Please find below our detailed responses to each of the points raised.

-----------------------------------------------------------------------------------------------------------------

  1. The abstract is not clear and needs to be rewritten.

Response: Many thanks for this comment. The abstract has been edited again showing the rationale of this research study and the specific objectives of the research. We also showed the main results obtained particularly the importance of using ground-based and satellite based remotely sensed data in detecting maize traits.

-----------------------------------------------------------------------------------------------------------------    

  1. 2. The traits and characteristics of plants have two different meanings. This manuscript is actually a remote sensing inversion of some maize indices. 

Response: We are agree with you. The term characteristic has been replaced by trait in the revised manuscript.    

-----------------------------------------------------------------------------------------------------------------

  1. Results and Discussion" is followed by "5. Conclusions"? I suggest writing results and discussion separately for more clarity.

Response: Thanks Sir. The numbering has been changed as it should be. We write results and discussion together because we tested different objectives in this study so it was difficult to write then separately. These objectives are to (1) assess the effects of different stress conditions on three measured traits (LAI, BFW and Chlm) of maize; (2) evaluate the performance of classification algorithms to map different crop types within the study area; (3) evaluate the efficiency of radiometric ground-based data and QuickBird satellite imagery based on vegetation-SRIs to assess the three measured traits of maize across full irrigation, water stress and salinity stress conditions; and (4) evaluate the performance of PLSR and MLR models based on vegetation-SRI-derived radiometric ground-based data and QuickBird satellite imagery and their combination to predict the three measured traits of maize. For example the aim of classifying images is totally different from the other objectives. Therefore, the results and discussion of each objective were combined and presented and discussed in details under different subtitles in the results and discussion section.

-----------------------------------------------------------------------------------------------------------------

  1. Two methods are used to classify maize. I think the difference between the selected species and maize is very big, and the plants with less difference from maize in the area should be selected.

Response: We are agree with you since different classes of the image are different but in the processing of the QuickBird image we worked to mask the processed image to just maize crops. Therefore, other classes were not used for the calculation of SRIs. 

-----------------------------------------------------------------------------------------------------------------  

  1. What do the letters a, b, c, and d after the numbers in Table 3 mean?

Response: It means that: the mean values of each maize trait with the same letter are not statistically different (p > 0.05) between different treatments among different fields. Duncan’s test was performed to say the investigated parameters are statistically different or not.  It was added under Table 3.

-----------------------------------------------------------------------------------------------------------------

I hope the explanation given above adequately addresses all reviewers’ comments. I would appreciate if the revised version of our manuscript would be considered for publication in Sensors.

Yours

Ebrahem M. Eid

[Kafrelsheikh University]

[Botany Department, Faculty of Science, Kafrelsheikh University, Kafr El-Sheikh 33516, Egypt]

[Phone number: 002010 22648840]

[Email address: ebrahem.eid@sci.kfs.edu.eg]

Reviewer 3 Report

In this paper, the radiometric ground-based data and Quickbird imagery were used to assess the leaf area index (LAI), biomass fresh weight (BFW) and chlorophyll meter (Chlm) of maize across full irrigation, water and salinity stress. The PLSR and MLR methods were used to estimate the three measured traits on basis of vegetation-SRIs. The results and analysis were showed in this paper in detail.

Main considerations about the study are shown: 

(1)    Quickbird images are used for the vegetation-SRIs calculation, therefore the radiometric correction is important. The images have been geometrically corrected and atmospheric corrected. Why not considering the BRDF correction to compensate for the solar altitude effect? Because it is important for the vegetation extraction. 

(2)    The spelling and format mistakes should be revised, such as “..Table 1” (line 194, line 208); 3.3. Evlation…..in the line 280.

Author Response

  1. May 2021

Dr. Vittorio M.N. Passaro 

Editor-in-Chief

Sensors

Dear Prof. Passaro,

Please find attached the revised manuscript titled ‘Integration of radiometric ground-based data and high resolution QuickBird imagery with Multivariate modeling to estimate maize traits in the Nile Delta of Egypt’. Manuscript ID: sensors-1209459, authored by Adel El-Metwalli, Andrew N. Tyler, Farahat S. Moghanm, Saad A.M. Alamri, Ebrahem M. Eid, and Salah Elsayed.

On behalf of my co-authors, I thank you very much for giving us the opportunity to revise our manuscript. We appreciate the positive and constructive comments and suggestions provided by the reviewers on our manuscript. We have carefully studied the reviewers’ comments and have made revisions that are indicated using track changes in the revised version of the manuscript. We have tried our best to revise our manuscript according to the reviewers’ comments. Please find attached the revised version of our manuscript, which we would like to submit for your kind consideration. Once again, we would like to express our great appreciation to you and the reviewers for the comments on our manuscript.Please find below our detailed responses to each of the points raised.

-----------------------------------------------------------------------------------------------------------------

(1) Quickbird images are used for the vegetation-SRIs calculation, therefore the radiometric correction is important. The images have been geometrically corrected and atmospheric corrected. Why not considering the BRDF correction to compensate for the solar altitude effect? Because it is important for the vegetation extraction. 

Response: Many thanks for this comment. The image was radiometrically corrected by the supplier of the image (Infoterra group based in the UK).

-----------------------------------------------------------------------------------------------------------------
(2) The spelling and format mistakes should be revised, such as “..Table 1” (line 194, line 208); 3.3. Evlation…..in the line 280.

Response: Sorry for these mistakes. The format of journal allows to wright Table started by capital letter. Evaluation in the line 280 was corrected. And the spelling was revised in the manuscript.

-----------------------------------------------------------------------------------------------------------------

I hope the explanation given above adequately addresses all reviewers’ comments. I would appreciate if the revised version of our manuscript would be considered for publication in Sensors.

Yours

Ebrahem M. Eid

[Kafrelsheikh University]

[Botany Department, Faculty of Science, Kafrelsheikh University, Kafr El-Sheikh 33516, Egypt]

[Phone number: 002010 22648840]

[Email address: ebrahem.eid@sci.kfs.edu.eg]

Round 2

Reviewer 2 Report

Thanks to the authors for their efforts to revise the manuscript. It is recommended to publish this manuscript.